# Hydrogel-Containing Biogenic Silver Nanoparticles: Antibacterial Action, Evaluation of Wound Healing, and Bioaccumulation in Wistar Rats

**DOI:** 10.3390/microorganisms11071815

**Published:** 2023-07-15

**Authors:** Sara Scandorieiro, Angela Hitomi Kimura, Larissa Ciappina de Camargo, Marcelly Chue Gonçalves, João Vinícius Honório da Silva, Wagner Ezequiel Risso, Fábio Goulart de Andrade, Cássia Thaïs Bussamra Vieira Zaia, Audrey Alesandra Stinghen Garcia Lonni, Claudia Bueno dos Reis Martinez, Nelson Durán, Gerson Nakazato, Renata Katsuko Takayama Kobayashi

**Affiliations:** 1Laboratory of Basic and Applied Bacteriology, Department of Microbiology, Center of Biological Sciences, State University of Londrina, Londrina 86057-970, Brazil;sarascandorieiromicro@gmail.com (S.S.); angela.hkimura@uel.br (A.H.K.); larissa.ciappina@gmail.com (L.C.d.C.); marcellychue@gmail.com (M.C.G.); gnakazato@uel.br (G.N.); 2Laboratory of Innovation and Cosmeceutical Technology, Department of Pharmaceutical Sciences, Center of Health Sciences, University Hospital of Londrina, Londrina 86038-350, Brazil; audrey@uel.br; 3Laboratory of Histopathological Analysis, Department of Physiological Sciences, Center of Biological Sciences, State University of Londrina, Londrina 86057-970, Brazil; joaovhs713@gmail.com (J.V.H.d.S.); andrade@uel.br (F.G.d.A.); 4Laboratory of Animal Ecophysiology, Department of Physiological Sciences, Center of Biological Sciences, State University of Londrina, Londrina 86057-970, Brazil; wagner@uel.br (W.E.R.); cbueno@uel.br (C.B.d.R.M.); 5Laboratory of Neuroendocrine Physiology and Metabolism, Department of Physiological Sciences, Center of Biological Sciences, State University of Londrina, Londrina 86057-970, Brazil; ctbvzaia@uel.br; 6Institute of Biology, State University of Campinas, Campinas 13083-862, Brazil; nelsonduran1942@gmail.com

**Keywords:** *Staphylococcus aureus*, *Pseudomonas aeruginosa*, *Escherichia coli*, green nanotechnology, *Fusarium oxysporum*, cicatrization, histological analysis, silver accumulation, in vivo

## Abstract

Wound infections are feared complications due to their potential to increase healthcare costs and cause mortality since multidrug-resistant bacteria reduce treatment options. This study reports the development of a carbomer hydrogel containing biogenic silver nanoparticles (bioAgNPs) and its effectiveness in wound treatment. This hydrogel showed in vitro bactericidal activity after 2 h, according to the time–kill assay. It also reduced bacterial contamination in rat wounds without impairing their healing since the hydrogel hydrophilic groups provided hydration for the injured skin. The high number of inflammatory cells in the first days of the skin lesion and the greater degree of neovascularization one week after wound onset showed that the healing process occurred normally. Furthermore, the hydrogel-containing bioAgNPs did not cause toxic silver accumulation in the organs and blood of the rats. This study developed a bioAgNP hydrogel for the treatment of wounds; it has a potent antimicrobial action without interfering with cicatrization or causing silver bioaccumulation. This formulation is effective against bacteria that commonly cause wound infections, such as *Pseudomonas aeruginosa* and *Staphylococcus aureus*, and for which new antimicrobials are urgently needed, according to the World Health Organization’s warning.

## 1. Introduction

The skin, an organ of great importance, provides a barrier between the internal and external environments of the body, protecting the organism against pathogens [1]. However, when this barrier is compromised, as observed in some wounds, the organism becomes more susceptible to infections [2,3]. A wound is defined as an impairment to the integrity of biological tissue, encompassing the skin and mucous membranes, caused by several types of traumas or surgery [4]. Wounds can be classified into two categories. Closed wounds can be caused by a contusion or hematoma, while open wounds may result from incisions, lacerations, punctures, penetrations, and thermal, chemical, or electrical burns [5]. The main complication associated with open wounds is the high risk of infection, which, when related to the patient’s clinical history, can pose substantial challenges to treatment efficacy and potentially lead to grave medical conditions [4,6].

The development of a clinical infection depends on various factors, including the pathogenicity of the microorganism, the immunological status of the host, and the condition of the wound [4]. However, within the wound, bacteria engage in competition with normal cells for oxygen and nutrients, while the presence of endotoxins and metalloproteinases can induce damage across all stages of wound healing [7]. Consequently, these factors contribute to prolonged hospital stays and increased healthcare costs [8]. Topical administration of antibiotics plays a crucial role in managing wound infections, as it promotes infection control and facilitates the healing process [9]. However, multidrug-resistant bacteria significantly limit treatment options and represent a major obstacle to effective infection management [10]. Notably, microbial resistance to silver sulfadiazine, a topical antibiotic commonly employed for treating wound infections, has already been reported [11,12,13]. Without intervention, projections suggest that by 2050, multidrug-resistant infections will result in one death every three seconds, amounting to 10 million deaths annually [14], surpassing the mortality rate caused by the COVID-19 pandemic. The recent pandemic has reinforced the urgency of combating bacterial resistance, given that respiratory viral diseases predispose patients to secondary infections involving pathogenic microorganisms [15,16].

According to the World Health Organization, there is an urgent need for the development of novel antimicrobials targeting twelve pathogenic bacteria that pose a significant threat to public health. This includes bacterial species implicated in wound infections, such as *Pseudomonas aeruginosa*, *Staphylococcus aureus,* and *Escherichia coli* [17]. Among the potential alternatives, nanotechnology-based antimicrobials have drawn considerable attention [18,19]. Previous studies conducted by our research group showed the bactericidal effects of silver nanoparticles biologically synthesized (bioAgNPs) with *Fusarium oxysporum* components. These bioAgNPs have exhibited efficacy against both Gram-positive and Gram-negative bacteria [20,21], including multidrug-resistant strains and bacteria associated with biofilm formation [22].

Silver nanoparticles have emerged as notable nanometals due to their antimicrobial activity against a wide range of bacteria. Moreover, they have been tested in combination with other antimicrobial agents to reduce toxicity and combat the emergence of bacterial resistance [23]. Firstly, the attractiveness of silver nanoparticles stems from their broad spectrum of antimicrobial action, allowing their incorporation into various everyday products [24,25]. Among metals, silver possesses a higher tendency for reduction (with a reduction potential of 0.80 V) compared to metals such as Cu (0.34 V), Fe (−0.44 V), and Zn (−0.76 V) [26]. The reduction potentials of these metals also contribute to their biological activities, as they facilitate the binding to electron-donating molecules within microorganisms, ultimately disrupting their cellular functions [27]. Nonetheless, it is crucial to highlight that factors such as size, morphology, zeta potential, and composition also exert substantial influence over the antimicrobial properties of nanometals [24].

Several commercially available topical treatment products, including dressings and ointments, incorporate chemically synthesized AgNPs into their composition [25]. However, the synthetic route employed in their production often involves toxic reducing agents and stabilizers [24]. In contrast, bioAgNPs offer a sustainable alternative as they are produced through a “green nanotechnology” approach that eliminates the need for chemical reagents, relying instead on the inherent reducing and stabilizing properties of biological entities [24]. In addition, biomolecules contribute to the improved stability of silver nanoparticles by preventing their aggregation [28]. Notably, the biological synthesis of silver nanoparticles, particularly bioAgNPs, has undergone extensive research and validation, providing a solid foundation for their production process [24]. In our study, we employed bioAgNPs derived from *F. oxysporum*, whose synthesis mechanism [29] and antimicrobial action have been comprehensively investigated [20,22]. These findings are pivotal in justifying the incorporation of these nanoparticles into the hydrogel formulation presented in this study. Therefore, our research group proposes the development of a bioAgNP-containing hydrogel for wound treatment. This article outlines the formulation of the hydrogel, evaluates its antimicrobial activity in both in vitro and in vivo settings, examines its effect on wound healing, and investigates potential bioaccumulation in animal models. Considering the low toxicity of these bioAgNPs and the urgent need for innovative wound treatment products, our presented formulation holds substantial promise for industrial, economic, and healthcare applications.

## 2. Materials and Methods

### 2.1. Biosynthesis of Silver Nanoparticles

Nanoparticle biosynthesis was performed as described by Duran et al. [29]. The principle of synthesis involves the reduction of silver ions by the components of *Fusarium oxysporum*. The fungal strain 551 was used in this process, and was provided by the Laboratory of Molecular Genetics, ESALQ-USP, Piracicaba, São Paulo, Brazil. The fungus was cultivated for seven days at 28 °C in a culture medium consisting of 2% (*w*/*v*) malt agar extract (Acumedia, San Bernardino, CA, USA), 0.5% (*w*/*v*) yeast extract (Becton, Dickinson and Company, Franklin Lakes, NJ, USA), and distilled water. The fungal mycelium was then added to distilled water at 0.1 g/mL and incubated at 28 °C for 72 h under agitation (150 rpm). Then, the solid components (mycelium and hyphal fragments) were separated by vacuum filtration using a qualitative filter with pore sizes ranging from 4 to 12 mm (Unifil). A 0.01 M solution of silver nitrate (AgNO_3_) was added to the fungal cell-free filtrate to synthesize silver nanoparticles. The system containing silver salt and fungal components was incubated statically for 15 days at 28 °C in the absence of light. To confirm nanoparticles synthesis, aliquots from the system were periodically collected to measure the absorption spectra in the range from 340 to 700 nm, using an ultraviolet-visible spectrophotometer (Thermo Scientific™ Multiskan™ GO Microplate Spectrophotometer, Marsiling Industrial Estate, Singapore), thereby verifying the plasmonic band of bioAgNPs. Additionally, the diameter and zeta potential of the silver nanoparticles were determined by photon correlation spectroscopy by Zetasizer NanoZS (Malvern^®^, Malvern, UK). The diameter measurement was determined under the following conditions: material with a refractive index (RI) of 0.2, absorption of 0.4; water as dispersant with an RI of 1.33, viscosity of 0.8872 cP; system at 25 °C, with a count rate of 450.5 kcps, duration of 60 s, measurement position of 5.5 mm, and attenuator of 7. Similarly, the zeta potential was determined under the following conditions: material and dispersant RI as described before; dispersant dielectric constant of 78.5; system at 25 °C, count rate of 111.5 kcps, zeta runs of 12, measurement position of 2 mm, and attenuator of 9. Finally, the morphology of bioAgNPs was determined by transmission electron microscopy (TEM).

### 2.2. Preparation of Hydrogels

Three hydrogel formulations were prepared as follows: a hydrogel without bioAgNPs (BASE), a hydrogel containing bioAgNPs at 250 μM (F250), and a hydrogel containing bioAgNPs at 500 μM (F500). The excipients used in the base formulation (negative control) included carbomer as the gelling agent, aminomethyl propanol as the neutralizing agent, glycerin as the humectant, and distilled water.

To prepare the formulations, the carbomer gelling agent was dispersed in distilled water under constant mechanical stirring (750 rpm) to obtain a homogeneous mixture. The neutralizing agent, aminomethyl propanol, was added, and the mixture was stirred until a gel consistency was achieved. Glycerin was then added to the gel under continued agitation, and finally, the bioAgNPs (250 μM or 500 μM), producing the F250 and F500 formulations. The pH of the three formulations (BASE, F250, and F500) was adjusted (using sodium hydroxide) between 5.5 and 6.5, which corresponds to the optimal pH value of the skin [30]. This formulation is patented (patent deposit was made in 2019, whose registration number is BR 102019003123-9 A2; http://www.inpi.gov.br (accessed on 28 May 2023)).

### 2.3. In Vitro Antibacterial Activity of Formulations

#### 2.3.1. Bacterial Samples

The in vitro antibacterial activity was tested against the following reference strains: *Escherichia coli* ATCC 25922, *Pseudomonas aeruginosa* ATCC 9027, and *Staphylococcus aureus* ATCC 25923. Bacterial strains were stored at −20 °C in BHI (Brain Heart Infusion, Acumedia^®^, San Bernardino, CA, USA) broth supplemented with 20% glycerol. For reactivation, the bacterial samples were cultured in BHI agar at 37 °C for 18 h.

#### 2.3.2. Time–kill Assay

Time–kill assay was performed to assess the antibacterial effect of three hydrogel formulations (BASE, F250, and F500) against the aforementioned reference strains. Two treatment controls were used: a 0.9% *w*/*v* saline solution as the negative control (CTL) and silver sulfadiazine (SS) at 1% *w*/*w* as a reference antimicrobial for wound treatment (AZICERIO^®^, Taquaritinga, São Paulo, Brazil).

To evaluate the antibacterial activity, 500 mg of each product (BASE, F250, F500, or SS) was added to a 100 mm diameter Petri dish, while a similar Petri dish received 500 μL of physiological solution as a control. The product or saline solution was uniformly spread across the entire surface of the Petri dish using a Drigalski spatula to form a homogeneous layer. Subsequently, to prepare the bacterial inoculum for the susceptibility test, isolated colonies grown on nutrient agar (HiMedia^®,^ Mumbai, Maharashtra, India) medium were suspended in 0.1 M PBS (Phosphate Buffer Solution, pH 7.2) to achieve turbidity equivalent to 0.5 McFarland scale, corresponding to 1.5 × 10^8^ CFU/mL. The bacterial suspension was diluted 1:10 in PBS to obtain an inoculum of 10^7^ CFU/mL. Then, 100 μL of bacteria at 10^7^ CFU/mL were inoculated onto all Petri dishes already containing hydrogel, commercial antimicrobial, or saline solution (negative control) using a Drigalski spatula. At five time points (0 h, 30 min, 2 h, 8 h, and 24 h) of treatment at 37 °C, Petri dishes were washed with 1 mL of PBS to recover surviving bacteria. The recovered bacterial load was serially diluted in PBS and subcultured on nutrient agar to determine CFU/mL.

### 2.4. In Vivo Study of Formulations

#### 2.4.1. General Design of In Vivo Experiment

Sixty male rats (*Rattus norvegicus albinus*) of Wistar lineage, with an average weight of 250 g, were utilized in this study. They were obtained from the Central Vivarium of the State University of Londrina (UEL). The animals were housed in a temperature-controlled environment (22 °C ± 2 °C) and exposed to a 12/12 h light/dark cycle. Access to potable water and food was provided ad libitum. The present study was approved by the Ethics Committee on Animals Usage from UEL, with protocol number 22730.2015.98.

The animals were randomly separated into five groups, each consisting of 12 animals. Four treatment courses were determined, and therefore the animals were further divided into four subgroups, with three rats each. Accordingly, each subgroup was sacrificed according to the specific treatment course (1st day, 5th day, 9th day, and 17th day) [31,32]. The five experimental groups were as follows: CTL (negative control—treatment with saline solution at 0.9% *w*/*v*); SS (silver sulfadiazine—AZICERIO^®^); BASE (hydrogel without bioAgNPs); F250 (hydrogel containing bioAgNPs at 250 μM), and F500 (hydrogel containing bioAgNPs at 500 μM).

#### 2.4.2. Surgical Excision of Skin

The animals underwent a four-hour food suspension prior to the surgical procedures, although they had unrestricted access to water. The rats were weighed to determine the appropriate amount of anesthetic solution for each individual. Anesthesia was administered intraperitoneally using a combination of ketamine (45 mg/mL) and xylazine (7 mg/mL) at a dose of 0.16 mL/100 g of body weight. While under the effect of anesthetic, the animals were positioned in ventral decubitus on the surgical table and immobilized with elastic tape. Depilation of the upper back region (thoracic spine) was performed on an area of approximately 6 cm^2^ [31,32,33].

For the surgical procedure, sterile scalpel blades were used to excise 4 cm^2^ of skin from the center of the depilated area, exposing the dorsal muscular fascia. Hemostasis was achieved through compression using gauze [31,32,33].

#### 2.4.3. Treatment of Animals with Hydrogel Formulations

After the surgery, the animals were individually placed in cages. Topical formulations were applied to the area where the skin was excised. The treatment protocol was as follows: 500 μL of 0.9% saline solution (*v*/*v*) was applied to the CTL group (negative control), while 500 mg of the formulated products (BASE, F250, F500) were applied according to the experimental design. To alleviate pain, minimize discomfort, and reduce the stress to which the animals were subjected, paracetamol (200 mg/mL) was administered orally via gavage for five days at a dose of 200 mg/kg of body weight [31,32,33,34].

#### 2.4.4. Wound Colonization

On days 1, 5, 9, and 17 following the injury, microbiological sampling was carried out in each group to evaluate bacterial colonization of the wounds. For microbial collection, a sterile swab (Firstlab) moistened with 0.9% saline solution (*w*/*v*) was gently swiped over the wound area. The swab was then streaked onto BHI agar in a Petri dish, which was subsequently incubated at 37 °C for 24 h. After bacterial growth, CFU/mL was determined.

#### 2.4.5. Wound Healing Analysis

The skin lesions of the animals were photographed daily throughout the 17-day treatment period using a digital camera (Motic 5, Xiamen, China). The photographs were then submitted to the Motic Images Plus 2.0 software (Moticam (Motic, Xiamen, China)) to measure the area of the skin lesions and analyze the progression of wound healing.

#### 2.4.6. Animal Euthanasia

After each designated treatment period (1st day, 5th day, 9th day, and 17th day), as defined in the experimental design, all three animals from each subgroup were euthanized by decapitation. Upon decapitation, blood samples were promptly collected in heparin-containing tubes to quantify silver accumulation. Subsequently, samples from the liver, spleen, brain, and kidneys of animals belonging to the CTL, SS, F250, and F500 groups were extracted to determine the levels of silver accumulation.

#### 2.4.7. Histopathological Analysis

The collected organs were fixed in Bouin’s solution for 48 h, followed by dehydration and embedding of the organs. The embedded specimens were fixed in appropriate cassettes and prepared for sectioning using a microtome (CUT 5062 SLEE MAINZ^®^, Mainz, Rhineland-Palatinate, Germany) to obtain 7 μm thick slices. These slices were then mounted onto histological glass slides. Hematoxylin and eosin staining was performed on the slides. Subsequently, observations and histological analysis were performed using a Moticam image capture system (Motic, Xiamen, China) coupled with a photomicroscope (Motic, Xiamen, China).

Micrographs captured at a magnification of 100× and of 400× were used to assess the number of blood vessels and inflammatory cells respectively present in each section. During the image capture, quadrants with an area of 250 μm^2^ were demarcatedand used to count inflammatory cells per animal, excluding quadrants that contained blood vessels. A total area of 1000 μm^2^ was used to count blood vessels. The total number of inflammatory cells was counted using the “Cell Counter” plugin with ImageJ software, version 1.50i [35], with differentiation from other cell types based on nuclear characteristics.

#### 2.4.8. Silver Bioaccumulation

The liver, brain, spleen, kidney, testis, and blood cells were dried in an incubator at 60 °C and subsequently digested in a 5N nitric acid solution at 60 °C for 48 h. After digestion, the material was centrifuged at 1000 RCF for 20 minutes, and the resulting supernatant was used to determine the amount of silver [36] using Atomic Absorption (AAnalyst ™ 700, PerkinElmer^®^, Waltham, MA, USA).

### 2.5. Statistical Analysis

The statistical analysis was performed by one-way analysis of variance (ANOVA) followed by Tukey’s test. The analysis was performed using R software, version 3.1.0 (Foundation for Statistical Computing, Vienna, Austria). Values of *p* < 0.05 were considered statistically significant.

## 3. Results

### 3.1. Characteristics of Biogenic Silver Nanoparticles (bioAgNPs)

The color of the fungal-free solution changed from pale yellow to brownish upon the addition of AgNO_3._ Over a period of 15 days at 28 °C, the intensity of the brown color increased. The absorption peak corresponding to the plasmonic band of bioAgNPs was observed at 420 nm, indicating successful synthesis of the bioAgNPs. This was further confirmed by photon correlation spectroscopy and microscopy analysis, which confirmed nanoparticle formation. The average diameter of the bioAgNPs diameter and the zeta potential were 81.25 nm and −36.4 mV, respectively. Transmission electron microscopy analysis of the bioAgNPs revealed spherical nanoparticles (Figure 1A). The size distribution of the nanoparticles and their zeta potential are shown in Figure 1B and Figure 1C, respectively.

As observed in Appendix A (Figure A1), the color analysis of the carbomer hydrogel showed that the BASE formulation exhibited a transparent color, while the bioAgNP hydrogel exhibited a uniform translucent caramel color, suggesting a homogeneous distribution of bioAgNPs within the gel.

### 3.2. Time–kill Assay (In Vitro Antibacterial Test)

For each bioAgNP formulation (F250 and F500) and the reference antimicrobial product (SS), a comparative analysis was conducted among four treatment time points (0 h, 2 h, 8 h, and 24 h), revealing statistically significant differences in terms of CFU/mL (*p* < 0.05, Tukey test), as presented below. According to the time–kill assay, F250, F500, and SS showed bactericidal activity against all tested bacterial species.

Regarding *E. coli* ATCC 25922 (Figure 2A) and *P. aeruginosa* ATCC 9027 (Figure 2B), the F500 formulation and SS reduced bacterial population by approximately 6 log (*p* < 0.05) after 8 h of treatment. The F250 formulation caused a reduction of approximately 6 log (*p* < 0.05) after 24 h of treatment. For *S. aureus* ATCC 25923 (Figure 2C), SS led to a reduction in bacterial population of about 6 log (*p* < 0.05) after 2 h of treatment, while the F250 and F500 formulations caused an approximately 6 log (*p* < 0.05) reduction after 8 h of treatment.

All tested bacteria (*E. coli*, *P. aeruginosa*, and *S. aureus)* exhibited similar population viability (*p* > 0.05) when exposed to PBS throughout the entire experiment, during all experiments, including the 24 h time point, thus serving as the growth control (CTL). As for the control of the hydrogel formulation, BASE did not significantly reduce the number of viable bacterial cells (*p* > 0.05) compared to the untreated CTL, as BASE lacks antimicrobial activity.

### 3.3. Results of In Vivo Study

#### 3.3.1. Wound healing

Regarding the healing process, there was no significant difference (*p* > 0.05, Tukey test) in the area of the skin lesion among the BASE, F250, and F500 groups compared to the CTL and SS groups. Figure 3 provides a qualitative representation of the skin lesions in the five animal groups. The quantitative evaluation of wound healing is presented in Table 1, which indicates the area of the skin lesions. Figure 4 illustrates the progression of wound recovery over the 17 days of treatment. F250 and F500 formulations demonstrated a similar reduction in wound area compared to SS treatment. F250, F500, and SS groups presented a significant reduction from 3.65 cm^2^ to 0.07 cm^2^ (98.08%), from 4.94 cm^2^ to 0.20 cm^2^ (95.95%), and from 4.32 cm^2^ to 0.21 cm^2^ (95.14%), respectively. The reduction in wound area observed in the BASE group was from 4.97 cm^2^ to 0.26 cm^2^ (94.77%). The CTL group displayed a reduction in wound area from 3.69 cm^2^ to 0.20 cm^2^ (94.58%).

#### 3.3.2. Quantification of Bacterial Cells in Wounds

Starting from the first day of skin injury, bacterial cultures of the wounds showed microbial growth in the CTL, SS, BASE, F250, and F500 groups (Figure 5). In the CTL group, the bacterial load was approximately 10^4^ CFU/mL at the beginning of the treatment with saline solution (after one or five days), and it increased to around 1 log after nine days. For the SS group, the initial bacterial count was approximately 10^2^ CFU/mL, which progressively increased in the following days, reaching more than 10^4^ CFU/mL after 17 days of treatment. In the F250 group, the bacterial load was approximately 10^4^ CFU/mL at the start of treatment, and it decreased to 10^2^ CFU/mL after 17 days. However, the F500 group exhibited a bacterial count of approximately 10^3^ CFU/mL at the beginning of the treatment, which decreased to less than 10^2^ CFU/mL after the 9th and 17th days.

#### 3.3.3. Histopathological Analysis

Regarding the histopathological analysis (Figure 6), on the first day after injury and treatment, all groups showed a predominance of inflammatory cells and few blood vessels. On the fifth day, there was still a dominance of inflammatory cells, and the number of blood vessels began to increase. On the ninth day, the inflammatory cell count decreased while the number of blood vessels increased in all groups. By the 17th day, there was a reduction in both inflammatory cells and blood vessels. The quantification of the cicatrizing effects revealed increased vascular proliferation on the fifth day, a peak in blood vessel formation on the ninth day, and an increased inflammatory infiltrate on the first and fifth days, which decreased in the subsequent days. No significant difference was observed among the groups (*p* > 0.05, Tukey test) in terms of blood vessels and inflammatory cells on the 1st, 5th, 9th, and 17th days. Figure 7 exemplifies the histological analysis, presenting a histological micrograph depicting the layers of rat skin, blood vessels, and inflammatory cells.

#### 3.3.4. Silver Quantification of Tissues

Silver quantification was performed in the CTL, SS, F250, and F500 groups after 17 days of treatment (Figure 8). None of the groups showed significantly higher concentrations of silver compared to CTL (p > 0.05, Tukey test). In the CTL group, the highest amount of silver was detected in the kidney, followed by the liver, spleen, brain, testis, and blood, ranging from 0.228 to 0.024 μg of silver per gram of tissue. For the SS group, the amount of silver ranged from 0.300 to 0.083 μg per gram of tissue, with the liver containing the highest amount of this metal. The F250-treated group showed a silver concentration ranging from 0.193 to 0.045 μg per gram of tissue, with the spleen exhibiting the highest accumulation of silver, followed by the liver, brain, testis, kidney, and blood. In the F500 group, the silver concentration in animals ranged from 0.177 to 0.061 μg of silver per gram of tissue after 17 days of treatment.

## 4. Discussion

Nanotechnology is a significant field of modern research that involves the design, synthesis, and manipulation of particle structures ranging approximately between 1 and 100 nm. Silver nanoparticles can be synthesized by chemical, physical, and biological methods. Chemical processes employ toxic chemicals that pollute the environment and are harmful to animals [37]. In contrast, physical methods may not require chemical agents but often need large physical workspaces and consume high amounts of energy [38]. Biological synthesis, however, offers an eco-friendly alternative by utilizing non-toxic molecules (e.g., amino acids, proteins, carbohydrates, etc.) derived from living organisms such as bacteria, filamentous fungi, yeasts, and plants [39].

This research proposes the incorporation of biologically synthesized silver nanoparticles into a hydrogel as an alternative to topical antibiotics for the treatment of infected wounds. Nowadays, silver has been used in topical pharmaceutical formulations such as silver sulfadiazine; however, this drug may cause adverse effects, exemplified by allergic reactions [40]. Conversely, advancements in nanoscience and nanotechnology have contributed to optimizing and functionalizing silver, thereby enhancing its antimicrobial activity and strategic mechanisms of action and reducing its toxicity, among other improvements [24]. For the development of the antimicrobial formulation in this study, biogenic silver nanoparticles synthesized using cell-free components of *F. oxysporum* were used as the active antimicrobial agent. 

The biosynthesis of *F. oxysporum*-bioAgNPs has been thoroughly characterized and validated. The nanoparticles from our research exhibited a plasmonic band at 420 nm, a spherical morphology (Figure 1A), an average size of 81.25 nm (Figure 1B), and a zeta potential of −36.4 mV (Figure 1C), which are consistent with the nanoparticles developed in previous studies [20,21,41,42]. Bocate et al. [42] reported bioAgNPs with a UV-Vis spectrum showing plasma absorption at 440 nm, a spherical shape confirmed by transmission electron microscopy (MET), an average size of 93 nm, and a zeta potential of −37.1 mV. In addition to supporting our study, we emphasize that these nanoparticles have already been characterized by X-ray diffraction (XRD) and Fourier-transform infrared spectroscopy (FTIR), showing silver patterns and characteristic bands of fungal proteins, respectively [42,43]. Furthermore, the biosynthesis mechanism was also investigated, involving the reduction of metal ions by nitrate-dependent reductase and an extracellular quinone shuttle process [29]. Since *F. oxysporum*-bioAgNPs have been well-characterized, their antimicrobial activity has been studied by our research group, showing efficacy against bacterial pathogens most commonly associated with infected wounds, such as *Staphylococcus aureus*, *Escherichia coli,* and *Pseudomonas aeruginosa* [20,21,44]. The exact mechanism by which silver nanoparticles exert their antibacterial activity is not yet fully understood. However, Scandorieiro et al. [20] showed that *F. oxysporum*-bioAgNPs cause damage to the bacterial cell wall and cytoplasmic membrane, as observed through electron microscopy and spectrometry, which revealed disruptions in these structures and leakage of cytoplasmic molecules, respectively. Additionally, the antibacterial effect of these nanoparticles is associated with oxidative stress, as evidenced by increased reactive oxygen species and lipid peroxidation. Other studies indicate that silver nanoparticles increase cell membrane permeability, inactivate enzymes, interfere with intracellular ATP levels, cause DNA damage, and induce the formation of reactive oxygen species [45,46,47,48,49].

Afterward, the present study confirmed the bactericidal effect of hydrogel containing bioAgNPs (F250 and F500) against *S. aureus, E. coli,* and *P. aeruginosa* through time–kill assays (Figure 2A,B and Figure 3B), which corroborates the findings of previous studies that tested the efficacy of this nanometal [50,51]. Our time–kill results showed that the bioAgNP hydrogel exhibited activity within 2 h of treatment, leading to a reduction in bacterial population after approximately 8 h. Thus, this formulation displayed a similar time of action to that of bioAgNPs alone, as observed in our earlier investigations of the mechanism of action of bioAgNPs [20]. Additionally, the time–kill data showed that SS significantly reduced the viability of the bacterial population. SS is a combination of sodium sulfadiazine and silver nitrate, with the silver ion binding to the nucleic acid of the microorganism and sulfadiazine interfering with microbial metabolism [52]. Although silver sulfadiazine has excellent antibacterial coverage, cases of bacterial resistance to this drug have been reported [53]. Furthermore, it can form a pseudo-eschar that delays wound healing [54]. Conversely, according to the time–kill assays, the BASE formulation did not cause a significant decrease in the viability of the bacterial population. Since BASE lacks any antimicrobial component, its effect resembled that of saline solution treatment. Nonetheless, the use of hydrogel in wound care is advantageous due to its hydrophilic functional groups, which facilitate water absorption, thereby promoting hydration and tissue recovery in the injured area [55,56]. According to the in vivo study presented in this article, BASE facilitated wound healing in a way comparable to other treatments, emphasizing the importance of adequate hydration for the wound-healing process [57]. 

As can be seen in the wound recovery data (Figure 3 and Figure 4), no significant difference was observed regarding wound healing among the Wistar rats treated with BASE, F250, and F500 when compared to the CTL and SS groups. These findings align with previous studies reporting the suitability of silver nanoparticles in dressings since they act as a barrier against pathogens, assist wound healing, and facilitate the removal of excess exudate. Moreover, silver nanoparticles are non-toxic and do not cause allergic reactions [58]. In addition, the efficacy of F250 and F500 against *P. aeruginosa*, *S. aureus,* and *E. coli,* as shown in Figure 2, was further confirmed in vivo. The wounds treated with these hydrogels showed a lower bacterial load compared to the CTL and SS groups. These data substantiate the role of the bioAgNP hydrogel in preventing wound contamination and colonization (Figure 5). Additionally, our findings corroborate the existing literature that reports the broad-spectrum action of silver nanoparticles [24], which probably contributed to the prevention of wound infection. We highlight that the bioAgNP hydrogel showed a homogeneous translucent caramel color, and all tested aliquots, both in vitro and in vivo, showed antimicrobial activity. This suggests that the distribution of nanoparticles within the hydrogel was adequate and likely homogeneous. However, it is worth emphasizing that further studies are required to characterize the formulation pharmacologically.

Later, we also analyzed wound healing by quantifying inflammatory cells and blood vessels, as illustrated in Figure 6. It is well-established that the wound-healing process consists of three main phases: inflammation, proliferation, and tissue remodeling [59,60]. Our results showed that the inflammatory process was evident in all groups (CTL, BASE, SS, F250, F500) on the first and fifth days of treatment. The high number of inflammatory cells observed is consistent with the initial stages of the wound-healing process, which are characterized by the inflammatory phase [32,61]. The inflammatory phase occurs immediately after a skin injury, accompanied by blood vessel rupture and the extravasation of their constituents. Initially, the local response aims to contain this extravasation, and inflammatory cells serve as the first line of defense of the body, being essential in the protection against pathogens. This cellular defense mechanism is prominent during the first week after injury and can persist if the wound becomes infected [62,63]. The subsequent proliferative phase involves the development of new blood vessels. Neovascularization is particularly important at this stage as it allows the transportation of oxygen and nutrients to the wound site [60,62,63]. Our results showed that the neovascularization process became evident on the fifth day post-injury, with peak angiogenesis occurring on the ninth day post-injury. As the wound progressed toward healing, angiogenesis gradually decreased, as indicated in Figure 6.

The benefits of silver nanoparticles in wound care are diverse, both in preventing infection and promoting the healing process. This nanometal has been incorporated into dressings, semi-permeable films, and semi-solid formulations [64]. However, most commercially available antimicrobial products containing silver employ ionic silver or nanoparticle forms synthesized by chemical processes involving toxic agents [25]. In contrast, our formulation comprises silver nanoparticles obtained through green nanotechnology, which avoids the use of chemical reagents and harnesses the reducing and stabilizing properties of *F. oxysporum* [29]. We emphasize that bioAgNPs present high stability due to biomolecules, which act as both reducing agents and efficient capping agents, preventing nanoparticle aggregation [28]. Furthermore, the hydrogel-bioAgNP formulation contains carbomer, facilitating its application and removal from wounds, thereby enhancing patient comfort. The pseudoplastic rheological behavior of this formulation favors optimal coverage of the wound area [65].

Another equally important analysis concerns the accumulation of silver in various organs and blood. As depicted in Figure 8, there was no significant difference between the CTL group and the experimental groups on the 17th day after treatment, indicating the absence of silver bioaccumulation related to SS and the bioAgNP hydrogel (F250 and F500). Nonetheless, the silver accumulation tended to be greater in the SS groups compared to the hydrogel groups containing bioAgNPs. This finding is consistent with the results obtained by Buzulukov et al. [66], who investigated intragastric administration of silver nanoparticles and observed bioaccumulation in the liver, kidney, spleen, heart, gonads, brain, and blood, with the highest metal content found in the liver. Despite the detection of this metal in various organs and tissues, the silver content was considered non-cytotoxic. Furthermore, biogenic silver nanoparticles generally exhibit lower in vivo toxicity when compared to nanoparticles synthesized through chemical routes or silver ions like those present in SS [67]. 

Finally, we highlight that the penetration of nanoparticles depends on their diameter. According to Larese et al. [68], particles larger than 45 nm cannot penetrate and permeate the skin. However, our hydrogel consisted of bioAgNPs with a larger size (81.25 nm). Although silver was detected in low quantities in the analyzed tissues and organs, this may be attributed to the presence of an open wound. It is worth noting that the literature reports that silver penetration can be up to five times greater in injured skin than in intact skin [69]. To the best of our knowledge, the present study is the first to report the quantification of silver in different organs of Wistar rats treated topically with a hydrogel containing bioAgNPs using atomic absorption spectrophotometry. Furthermore, there was no significant silver accumulation observed during the 17-day treatment period, as the silver content was similar in both groups of animals treated with and without the silver formulation. This lack of silver accumulation in the organs can be attributed to the gradual release of silver from the nanoparticles, as reported in the literature [70]. Additionally, the release of active substances can be defined as the process by which the active is released from the pharmaceutical formulation and becomes available for absorption [71].

In conclusion, the carbomer-based hydrogel containing bioAgNPs represents a promising antimicrobial alternative for wound infections, as it does not lead to silver bioaccumulation and does not interfere with wound healing. In addition, our findings revealed a significant increase in the number of inflammatory cells during the first days of injury, while the peak of neovascularization occurred one week after the onset of the wound. Further studies are needed to elucidate the in vivo mechanism underlying the antibacterial effect of bioAgNPs, and the development of novel formulations combining bioAgNPs with other antimicrobials is needed to address the challenges posed by emerging bacterial resistance.

## 5. Patents

This formulation has been patented. The patent deposit was made in 2019, and its registration number is BR 102019003123-9 A2; http://www.inpi.gov.br (accessed on 28 May 2023).

## Figures and Tables

**Figure 1 microorganisms-11-01815-f001:**
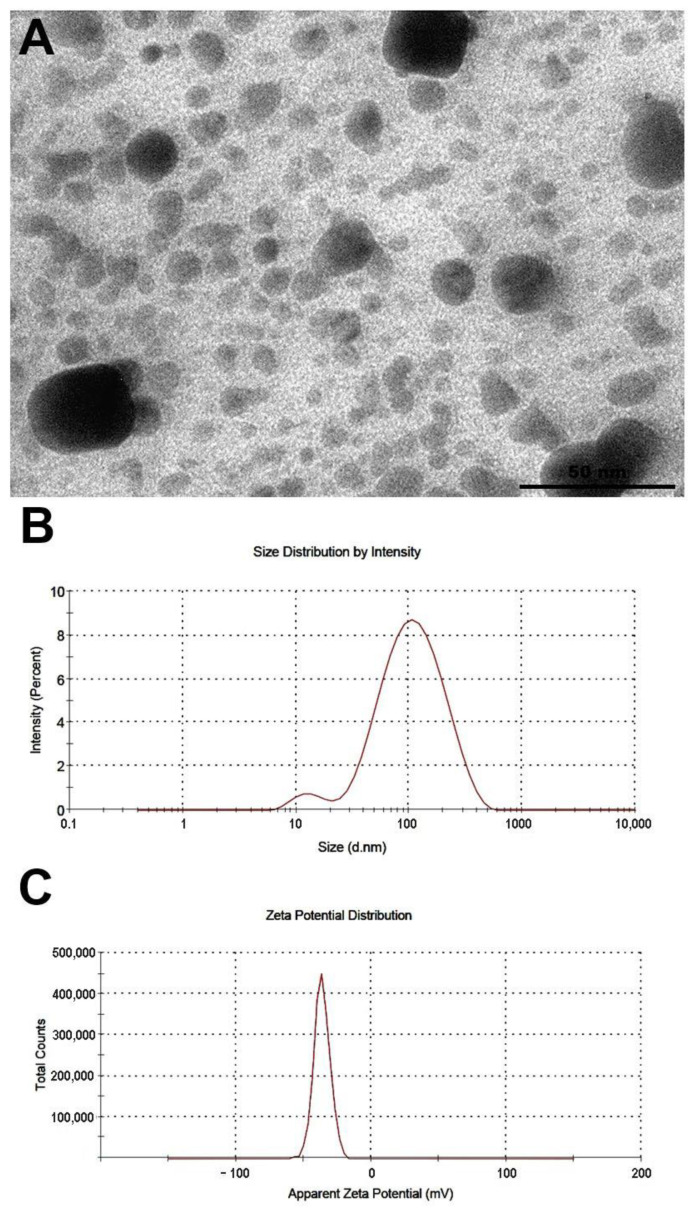
Characterization of the biogenic silver nanoparticles (bioAgNPs). (**A**) Transmission electron microscopy analysis of the bioAgNPs. Scale bar: 50 nm (**B**) Size distribution of the bioAgNPs was determined by intensity (%) and presented as a graph. The polidispersity index (PDI) was 0.296, indicating the uniformity of the nanoparticle size distribution. (**C**) Zeta potential distribution of the bioAgNPs.

**Figure 2 microorganisms-11-01815-f002:**
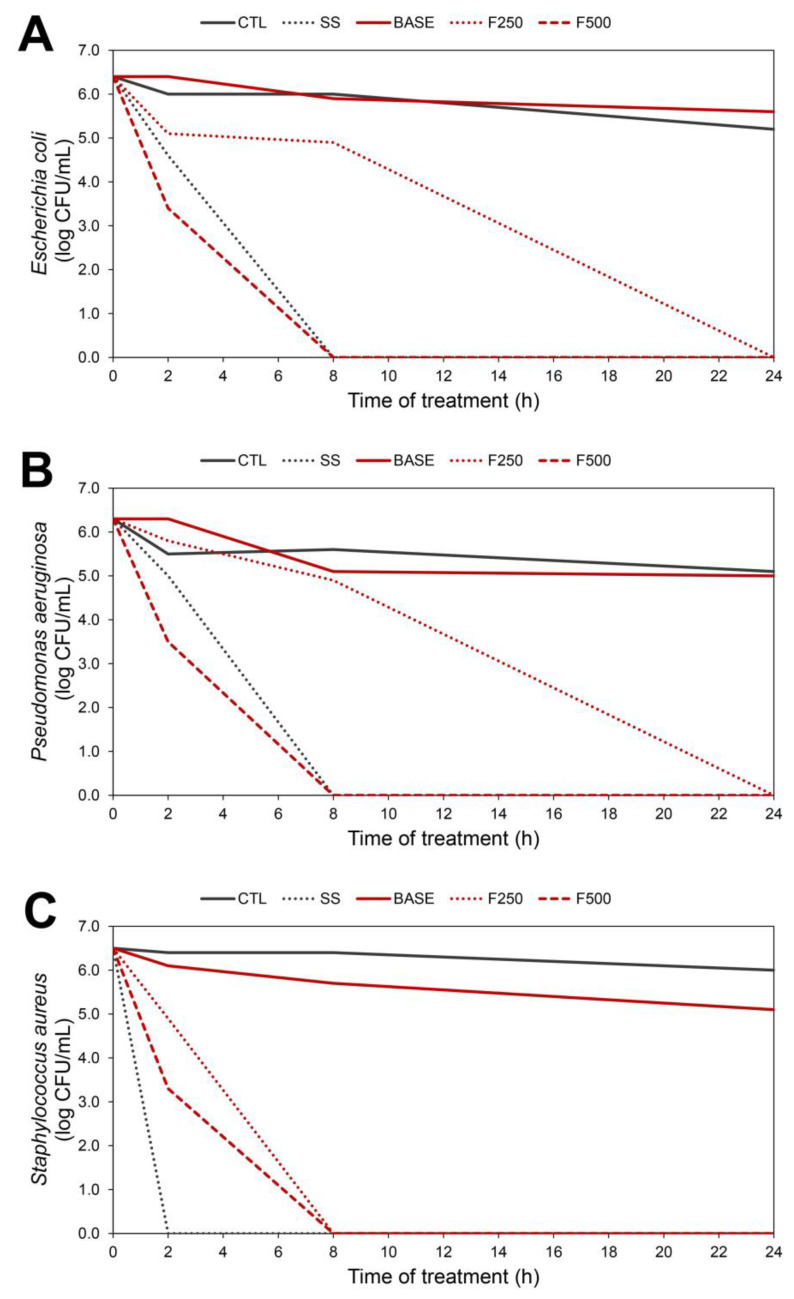
In vitro time–kill curve of bacterial species exposed to hydrogel formulations containing bioAgNPs at 250 μM (F250) and 500 μM (F500). Hydrogel without bioAgNPs displayed bacterial growth, indicating that BASE has no antimicrobial action. Control group (CTL) indicates the viability of the bacterial cells, as it does not contain any antimicrobial agent. Silver sulfadiazine (SS) was used as a reference antimicrobial because it is commonly used in wound treatment. Bacterial population is expressed in log_10_ colony-forming units (CFU)/mL. The results for each bacterial species are as follows: (**A**) *Escherichia coli* ATCC 25922, (**B**) *Pseudomonas aeruginosa* ATCC 9027, and (**C**) *Staphylococcus aureus* ATCC 25923.

**Figure 3 microorganisms-11-01815-f003:**
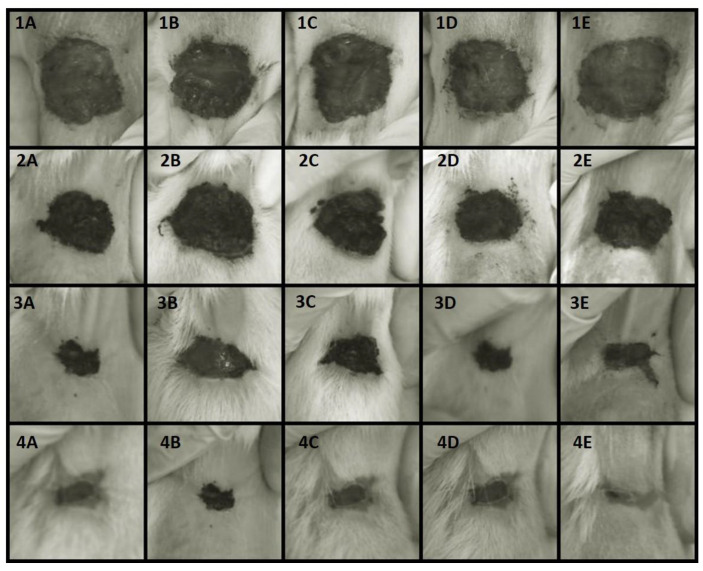
Evolution of wound healing in animals during a 17-day exposure to different treatments following a skin incision on the upper back. (**A**) Control group (CTL) treated with saline solution. (**B**) Silver sulfadiazine group (SS, antimicrobial reference treatment). (**C**) Non-bioAgNP hydrogel group (BASE). (**D**) The group treated with hydrogel containing bioAgNPs at 250 µM (F250). (**E**) The group treated with hydrogel containing bioAgNPs at 500 µM (F500). The treatment duration for each group was as follows: (1) 1 day, (2) 5 days, (3) 9 days, and (4) 17 days.

**Figure 4 microorganisms-11-01815-f004:**
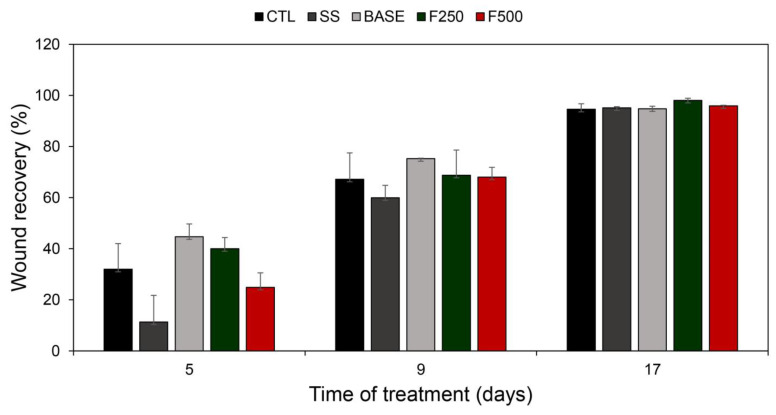
Recovery of the rat skin wound was assessed by comparing the initial lesion size with measurements taken at 5, 9, and 17 days after treatment with saline solution (CTL), silver sulfadiazine (SS), hydrogel with bioAgNPs at 250 µM (F250), and hydrogel with bioAgNPs at 500 µM (F500). The wound recovery is expressed as a percentage ± standard deviation.

**Figure 5 microorganisms-11-01815-f005:**
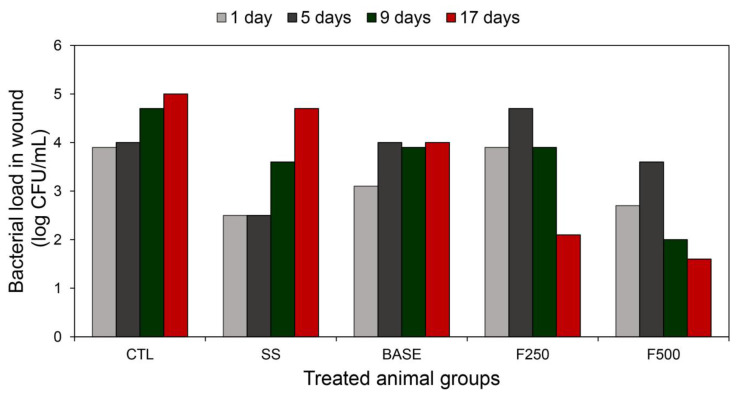
Bacterial load in the animal wounds was monitored on the 1st, 5th, 9th, and 17th days of treatment with saline solution (CTL), silver sulfadiazine (SS), hydrogel without bioAgNPs (BASE), hydrogel with bioAgNPs at 250 µM (F250), and hydrogel with bioAgNPs at 500 µM (F500).

**Figure 6 microorganisms-11-01815-f006:**
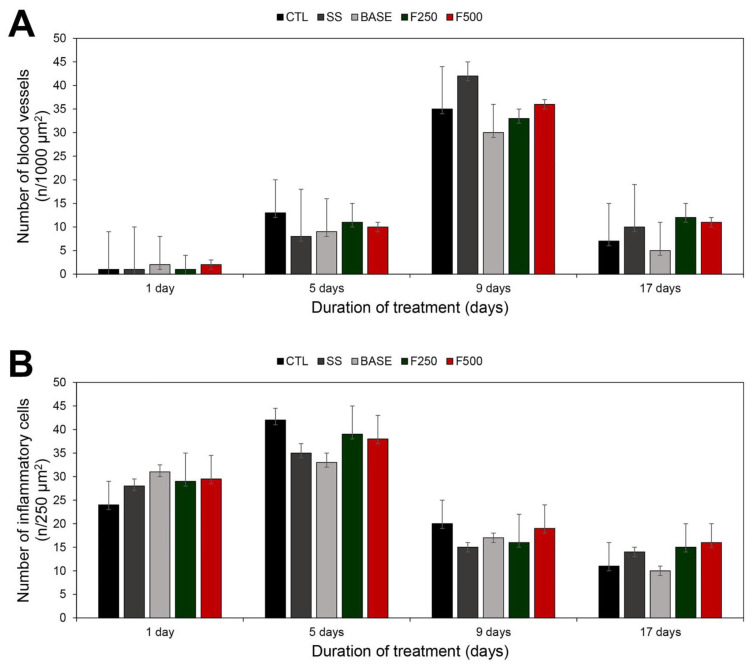
Quantification of blood vessels and inflammatory cells through micrograph analysis in animals treated with saline solution (CTL), silver sulfadiazine (SS), hydrogel without bioAgNPs (BASE), hydrogel with bioAgNPs at 250 µM (F250), and 500 µM (F500). The quantification of vessels and inflammatory cells was conducted after 1, 5, 9, and 17 days of treatment, and the results are presented as mean ± standard deviation. (**A**) Blood vessels were counted in 1000 μm^2^ quadrants of tissue sections. (**B**) Inflammatory cells were counted in 250 μm^2^ quadrants of tissue sections.

**Figure 7 microorganisms-11-01815-f007:**
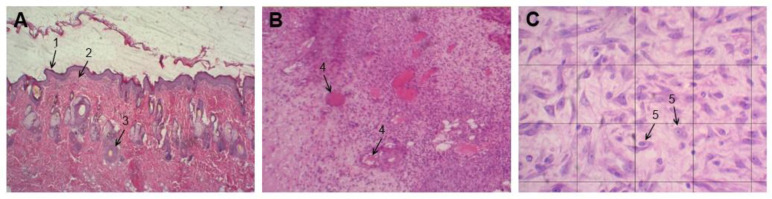
Histological micrographs stained with hematoxylin-eosin for the group treated with saline solution. (**A**) Micrograph taken at 40× magnification showing the epidermis (1), dermis (2), and blood vessels (3). (**B**) Micrograph taken at 100× magnification, used for counting blood vessels, indicated by number (4). (**C**) Micrograph taken at 400× magnification showing quadrants with an area of 250 μm^2^ each used for counting inflammatory cells, indicated by number (5). Observations and histological analysis were performed using a Moticam image capture system (Motic, Xiamen, China) coupled with a photo-microscope (Motic, Xiamen, China).

**Figure 8 microorganisms-11-01815-f008:**
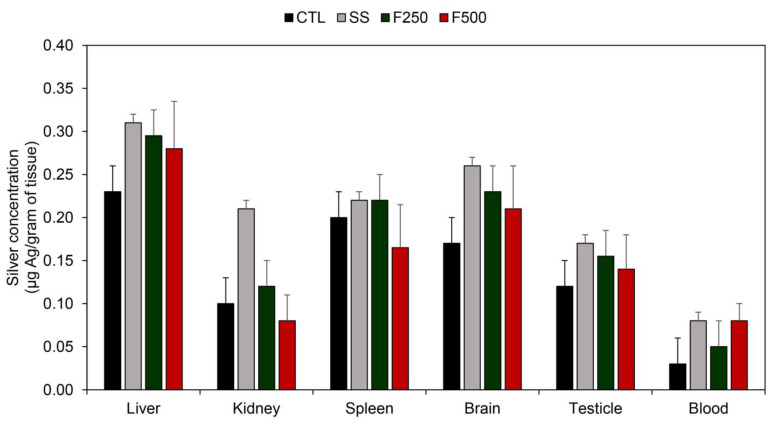
Silver quantification by atomic absorption spectrophotometry in several organs and tissues of animals treated with saline solution (CTL), silver sulfadiazine (SS), hydrogel with bioAgNPs at 250 µM (F250), and hydrogel with bioAgNPs at 500 µM (F500). The silver content is shown for the 1st and 17th day of treatment and is expressed in micrograms (µg) of silver per gram of tissue.

**Table 1 microorganisms-11-01815-t001:** Size of lesions in animals from the CTL, SS, BASE, F250, and F500 groups after different treatment durations (1, 5, 9, and 17 days). The lesion area values represent the mean ± standard deviation.

Groups	Lesion Area (cm^2^)			
	Day 1	Day 5	Day 9	Day 17
CTL	3.69 ± 0.61	2.51 ± 0.37	1.21 ± 0.38	0.20 ± 0.08
SS	4.32 ± 0.51	3.83 ± 0.45	1.73 ± 0.21	0.21 ± 0.02
BASE	4.97 ± 0.75	2.75 ± 0.25	1.23 ± 0.01	0.26 ± 0.05
F250	3.65 ± 0.14	2.19 ± 0.16	1.14 ± 0.36	0.07 ± 0.03
F500	4.94 ± 0.42	3.71 ± 0.28	1.58 ± 0.19	0.20 ± 0.01

CTL, control; SS, silver sulfadiazine; BASE, hydrogel without bioAgNPs; F250, hydrogel containing bioAgNPs at 250 µM; F500, hydrogel containing bioAgNPs at 500 µM; bioAgNPs, biogenic silver nanoparticles.

## Data Availability

Not applicable.

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
