# Peer review of "Hydrogel-Containing Biogenic Silver Nanoparticles: Antibacterial Action, Evaluation of Wound Healing, and Bioaccumulation in Wistar Rats"

_microorganisms, 2023, doi:10.3390/microorganisms11071815_

Round 1

Reviewer 1 Report

Comments for microorganisms-2452528

This article is good, meaningful, and clinically valuable. But in some places, there are small mistakes.

1. Line 37-38, for the first timemay be removed, because it is absolutely.

2. Line 50 and 55, There is no end to the sentence

3. In introduction,Several paragraphs can be combined into one or two paragraphs.

4. You can delete ")" in line or add "(" to line 91.

5. Line 98,“12 mmay be 12 mm.

6. In figure 1, A may small, B and C may be larger.

7. Figure 2,3,4 can be merged into a single graph.

8. The preface emphasizes a skin burn infection, while the study used a surgical removal of the skin wound. They are different.

9. In figure 7, the line indicated SD or SE and difference symbol should be presented.

10. In discussion, the literature is somewhat poorly cited.

11. In References, some of the literature lacks the DOI.

Author Response

Londrina, June 20th, 2023.

Dear Editor and Reviewers,

We did the modifications as suggested which are marked up using the “Track Changes” function of MS Word. We believe that the paper is better now. Thank you for your evaluation and suggestions.

Details of the revisions and the answers to the referees comment follow below.

Response do Reviewer 1:

This article is good, meaningful, and clinically valuable. But in some places, there are small mistakes.

  1. Line 37-38, “for the first time” may be removed, because it is absolutely.
  2. Line 50 and 55, There is no end to the sentence
  3. In introduction,Several paragraphs can be combined into one or two paragraphs.
  4. You can delete ")" in line or add "(" to line 91.
  5. Line 98,12 m”may be “12 mm”.
  6. In figure 1, A may small, B and C may be larger.
  7. Figure 2,3,4 can be merged into a single graph.
  8. The preface emphasizes a skin burn infection, while the study used a surgical removal of the skin wound. They are different.
  9. In figure 7, the line indicated SD or SE and difference symbol should be presented.
  10. In discussion, the literature is somewhat poorly cited.
  11. In References, some of the literature lacks the DOI.

About de comment:

“1.     Line 37-38, “for the first time”may be removed, because it is absolutely.

Thanks for the suggestion. We agreed with the reviewer and  made the change.

About the comment:

“2.     Line 50 and 55, There is no end to the sentence”

We added full stop to end both sentences of line 50 and line 55.

About the comment:

“3.     In introduction,Several paragraphs can be combined into one or two paragraphs.”

We have reorganized the “Introduction”, we combined paragraphs as suggested and added information about the topic of this study, meanly about wound infection and nanotechnology. Thanks for the suggestion, we believe the introduction section is better now.

About the comment:

“4.     You can delete ")" in line or add "(" to line 91.”

The reviewer is right, we corrected the typo and added “(“ in the mentioned part.

About the comment:

“5.     Line 98,12 m”may be “12 mm”.”

We corrected the typo in the unit of measure.

About the comment:

“6.     In figure 1, A may small, B and C may be larger.”

We agree with reviewer. We rearranged the images in figure 1 so that images “B” and “C” are larger and more readable.

About the comment:

“7.     Figure 2,3,4 can be merged into a single graph.”

We agree with the suggestion to join all time-kill data into a single figure. We merged the 3 time-kill graphics into figure 2.

About the comment:

“8.     The preface emphasizes a skin burn infection, while the study used a surgical removal of the skin wound. They are different.”

We are grateful for this observation. We agree with reviewer that our study did not evaluate burn-type injuries, but surgical-type open injuries. The intention of the present study was to investigate the antibiotic effect of the bioAgNP-hydrogel on open wounds that are extremely susceptible to infection. We have made changes to the abstract, introduction, and discussion sections so that it doesn't focus on burns, but on wounds. We also removed references that focused on burns, and added literature that focused on wounds, particularly open wounds.

About the comment:

“9.     In figure 7, the line indicated SD or SE and difference symbol should be presented.”

For clarity and to avoid confusion, we have changed the symbol used for silver sulfadiazine in all manuscript and figures, including figure 7. Previously the symbol for silver sulfadiazine was "SD" and now it is "SS”. Thanks for suggestion, we believe “SS” is more suitable.

About the comment:

“10.    In discussion, the literature is somewhat poorly cited.”

We restructured the discussion to be more in depth by exploring both our data and literature.  We improved the presentation of the nanoparticles that were produced and used in this work by highlighting their advantages towards silver nanoparticles obtained by chemical routes, their known biosynthesis mechanism, bioAgNP physicochemical characteristics, and their antimicrobial effect and mechanism of action. We presented that bioAgNP is validated by our research group, showing the importance of incorporating this well-characterized active in a hydrogel for the treatment of wound infections. The bioAgNP-hydrogel efficacy shown by in vivo test in this study is important data to allow this product to be evaluated in next step by clinical trials as alternative antimicrobial to combat microbial resistance.

About the comment:

“11.    In References, some of the literature lacks the DOI.”

As suggested, we have revised all references. The DOI was informed for all articles that had this identification available. Unfortunately, some of the articles do not have it informed, then we followed journal instructions that says the DOI must be added if available. In the guide to authors, whose link follows below, the following is explained: “Journal references must cite the full title of the paper, page range or article number, and digital object identifier (DOI) where available. “

https://www.mdpi.com/authors/references

Reviewer 2 Report

In this paper, the authors investigate a hydrogel with bioAgNP for the treatment of wounds. The results were useful for the researchers who worked in antibacterial materials and wound treatment materials. However, the synthesis process of silver nanoparticles and the structure and properties of gels are not researched. I have some major comments. If this manuscript can be satisfactorily revised according to my suggestions as below, I am willing to recommend this work to be published in Microorganisms.

Materials and methods

I suggest the authors pay attention to the antimicrobial mechanism of biosynthesis of silver nanoparticles in the fungal cell-free filtrate. For example, what is the change in the chemical structure of the components in the fungal cell-free filtrate during the synthesis of silver nanoparticles? 

Line 100-106, the characterization method of ultraviolet-visible spectrophotometry, diameter and zeta potential, the bioAgNP morphology should be described in sufficient detail to allow readers to replicate the experiment.

Are the particles uniformly dispersed in the hydrogel's network structure? The morphology of the gel can be observed via TEM by freeze-drying the hydrogel sample.

The bioAgNP releasing behavior of the hydrogel?

Author Response

Londrina, June 20th, 2023.

Dear Editor and Reviewers,

We did the modifications as suggested which are marked up using the “Track Changes” function of MS Word. We believe that the paper is better now. Thank you for your evaluation and suggestions.

Details of the revisions and the answers to the referees comment follow below.

Response do Reviewer 2:

In this paper, the authors investigate a hydrogel with bioAgNP for the treatment of wounds. The results were useful for the researchers who worked in antibacterial materials and wound treatment materials. However, the synthesis process of silver nanoparticles and the structure and properties of gels are not researched. I have some major comments. If this manuscript can be satisfactorily revised according to my suggestions as below, I am willing to recommend this work to be published in Microorganisms.

Materials and methods

I suggest the authors pay attention to the antimicrobial mechanism of biosynthesis of silver nanoparticles in the fungal cell-free filtrate. For example, what is the change in the chemical structure of the components in the fungal cell-free filtrate during the synthesis of silver nanoparticles?

Line 100-106, the characterization method of ultraviolet-visible spectrophotometry, diameter and zeta potential, the bioAgNP morphology should be described in sufficient detail to allow readers to replicate the experiment.

Are the particles uniformly dispersed in the hydrogel's network structure? The morphology of the gel can be observed via TEM by freeze-drying the hydrogel sample.

The bioAgNP releasing behavior of the hydrogel?

About the comment:

“Materials and methods

I suggest the authors pay attention to the antimicrobial mechanism of biosynthesis of silver nanoparticles in the fungal cell-free filtrate. For example, what is the change in the chemical structure of the components in the fungal cell-free filtrate during the synthesis of silver nanoparticles?”

We agree with the reviewer that the synthesis mechanism is relevant information in our study. Previously, the biosynthesis mechanism was investigated (Durán et al., 2005; http://www.jnanobiotechnology.com/content/3/1/8). In addition, our research group has characterized such nanoparticles regarding several parameters, including it has even been shown by infrared spectroscopy that the bioAgNP have fungal components which may be important for their stabilization. After the important question made by the reviewer, we citated in this manuscript the previous studies from our research group to support the validation of bioAgNP that justify the importance of incorporating them into a formulation and testing it in vivo. To make it clear that the biosynthesis and characteristics of bioAgNP are validated, we have added the following part to the discussion:

“The biosynthesis of F. oxysporum-bioAgNP is well characterized and validated, so that the plasmonic band at 420 nm, spherical morphology (figure 1A), size average (81.25 nm; figure 1A) and zeta potential (- 36.4 mV; figure 1C) of the nanoparticles from our re-search is similar to the nanoparticles developed by previous studies [20,21,41,42]. Bocate et al. [42] reported bio-AgNP whose UV–Vis spectrum showed the plasma absorption at 440 nm, spherical shape shown by MET, average size of 93 nm, and zeta potential of −37,1 mV. In addition to the results that collaborate with our study, we emphasize that these nanoparticles have already been characterized by X-ray diffraction (XRD) and Fourier transform infrared spectroscopy (FTIR), showing patterns of silver and characteristic bands of fungal proteins, respectively [42,43]. In addition, the mechanism of biosynthesis was also investigated, which involves reduction of metal ions by nitrate-dependent reductase and a shuttle quinone extracellular process [29]. Since F. oxysporum-bioAgNP are well-characterized, their antimicrobial activity has been studied by our research group; they showed activity against bacterial pathogens isolated from infected wounds, such as Staphylococcus aureus, Escherichia coli and Pseudomonas aeruginosa [20,21,44]. The mechanism by which silver nanoparticles exert their antibacterial activity is not yet fully under-stood; however, Scandorieiro et al. [20] showed that F. oxysporum-bioAgNP involves damage to the cell wall and cytoplasmic membrane, since disruption of these structures and cytoplasmic molecule leakage were identified by electron microscopy and spectrometry respectively. In addition, the antibacterial effect of these nanoparticles is due to oxidative stress confirmed by increased reactive oxygen species and lipid peroxidation. Other studies indicate that silver nanoparticles increase cell membrane permeability, inactivate enzymes, interfere with intracellular ATP levels, cause DNA damage and induce the formation of reactive oxygen species [45–49].”

About the comment:

“Line 100-106, the characterization method of ultraviolet-visible spectrophotometry, diameter and zeta potential, the bioAgNP morphology should be described in sufficient detail to allow readers to replicate the experiment.”

Thank you for pointing this out. As suggested, we added details to methodologies of characterization of nanoparticles.

Before the suggestion of reviewer, the description of the methodology was as follows: “Using ultraviolet-visible spectrophotometry (Thermo Scientific™ Multiskan™ GO Microplate Spectrophotometer), the synthesis of nanoparticles was accompanied by measurement of the absorption spectrum to verify the plasmonic band of bioAgNP. Lastly, the diameter and zeta potential of silver nanoparticles were determined by photon correlation spectroscopy by Zetasizer NanoZS (Malvern®). The bioAgNP morphology was determined by transmission electron microscopy (TEM).”

Now, after the change, the description of the methodology is as follows:

“To confirm nanoparticles synthesis, aliquots of the system were removed for measuring absorption spectra in range from 340 to 700 nm, using an using ultraviolet-visible spectrophotometry (Thermo Scientific™ Multiskan™ GO Microplate Spectrophotometer), to verify the plasmonic band of bioAgNP. Lastly, the diameter and zeta potential of silver nanoparticles were determined by photon correlation spectroscopy by Zetasizer NanoZS (Malvern®). The diameter was determined under the following conditions: material with refractive index (RI) of 0.2, absorption of 0.4; water as dispersant with RI of 1.33, viscosity of 0,8872 cP; system at 25°C, with count rate of 450.5 kcps, duration of 60 s, measurement position of 5.5 mm, and attenuator of 7. The zeta potential was determined under the following conditions: material and dispersant RI as described before; dispersant dielectric constant of 78.5; system at 25°C, count rate of 111.5 kcps, zeta runs of 12, measurement position of 2 mm, and attenuator of 9. The bioAgNP morphology was determined by transmission electron microscopy (TEM).”

About the comment:

“Are the particles uniformly dispersed in the hydrogel's network structure? The morphology of the gel can be observed via TEM by freeze-drying the hydrogel sample.”

Our study focused on testing the in vivo efficacy of bioAgNP incorporated into a hydrogel formulation, with the aim of verifying the antimicrobial effect, silver accumulation and interference in healing. Regarding the characteristics of the formulated hydrogel, there are indications that the silver nanoparticles were well dispersed in the hydrogel, because the base of the gel, which is transparent, acquired a homogeneous translucent caramel color after the addition of the nanoparticles. In addition, all aliquots (which were numerous), removed from the gel for microbiological tests, showed antimicrobial efficacy in vitro and in vivo, which indirectly confirms that the nanoparticles were homogeneously distributed in the gel. We added photography of BASE and bioAgNP-hydrogel to show their macroscopic appearance (appendix A, Figure 1A). The gel was not observed via TEM because the unique transmission electron microscope of university was not available because it was under repair. But we believe that the freeze-drying technique associated to MET can be used in future study focused on the detailed physical-chemical characterization of this formulation.

About the comment:

“The bioAgNP releasing behavior of the hydrogel?”

The aim of our study was to investigate the effect in vivo of bioAgNP in terms of antimicrobial efficacy, silver bioaccumulation and interference in wound healing. Since the formulation we developed showed efficacy to prevent wound infections without interfering in healing and without causing significant bioaccumulation, future studies are needed to investigate the characterization of this formulation that may be performed according to methodologies used previously by our co-author member (*Lonni et al., 2016; doi: 10.3109/10837450.2015.1081611). We emphasize that the characterization of this formulation, which had its in vivo efficacy confirmed here, requires a specific study, which will be carried out as the next step of this research, aiming to investigate the content of active, rheological properties of formulation, its in vitro release, skin permeation, stability among others.

*Lonni et al. (2015). Development and characterization of multiple emulsions for controlled release of Trichilia catigua (Catuaba) extract, Pharmaceutical Development and Technology, v. 21, n. 8.

However, we have added the following part to the discussion section to relate the bioaccumulation assay with literature permeation data:

“Finally, we highlight that the penetration of nanoparticles depends on their diameter. According to Larese et al. [68], particles larger than 45 nm cannot penetrate and permeate the skin, however, our hydrogel was composed of bioAgNP with larger size (81.25 nm). Silver was detected in low amounts in the analysed tissues and organs, probably due to the open wound, and in the literature, it is reported that silver penetration can be up to five times greater in injured skin than in intact skin [69]. According to our knowledge, the pre-sent study is the first report of silver quantification by atomic absorption spectrophotometry in different organs of Wistar rats treated with hydrogel containing biological silver nanoparticles. Furthermore, there was no silver accumulation during the 17 days of treatment, as the silver content was similar in both groups of animals treated with and without silver formulation.  Probably the absence of silver accumulation in the organs occurred because the silver from nanoparticles is gradually released, as reported in the lit-erature [70]; in addition, the release of actives can be defined in a simple way as the pro-cess that the active is released from the pharmaceutical form and becomes available to be absorbed [71].”

Reviewer 3 Report

The manuscript by Scandorieiro et al.Hydrogel containing biogenic silver nanoparticles: antibacterial action, evaluation of wound healing and bioaccumulation in Wistar rats” described a suitable approach for the synthesis of biogenic silver nanoparticles-based hydrogel for efficient wounds treatment. Overall, the manuscript is noteworthy and requires revision before its publication as follows:

Comments:

1.     Introduction, please add one paragraph on the advantages of nanoparticles (NPs) as antimicrobial agents, the suitability of hydrogel-based NPs synthesis and application or green synthesis approaches, how silver is suitable over other NPs such as gold, and the mechanism of NP’s action against pathogens, etc. RSC advances 6 (2016), 86808-86816; Biol Trace Elem Res 199 (2021) 2552–2564; Indian Journal of Microbiology 59 (2019), 379-382.

2.     Overall, the discussion section is weak and substantial improvement in the context of the quantitative data-based discussion to justify the significance of present finds with the citations.

3.     Please add one illustration to the summary of the present study along with the mechanism of antimicrobial action of biosynthesis NPs hydrogel.

4.     Few Figure’s quality should be improved.

Minor editing of English language is required.

Author Response

Londrina, June 20th, 2023.

Dear Editor and Reviewers,

We did the modifications as suggested which are marked up using the “Track Changes” function of MS Word. We believe that the paper is better now. Thank you for your evaluation and suggestions.

Details of the revisions and the answers to the referees comment follow below.

Response do Reviewer 3:

The manuscript by Scandorieiro et al. “Hydrogel containing biogenic silver nanoparticles: antibacterial action, evaluation of wound healing and bioaccumulation in Wistar rats” described a suitable approach for the synthesis of biogenic silver nanoparticles-based hydrogel for efficient wounds treatment. Overall, the manuscript is noteworthy and requires revision before its publication as follows:

Comments:

  1. Introduction, please add one paragraph on the advantages of nanoparticles (NPs) as antimicrobial agents, the suitability of hydrogel-based NPs synthesis and application or green synthesis approaches, how silver is suitable over other NPs such as gold, and the mechanism of NP’s action against pathogens, etc. RSC advances 6 (2016), 86808-86816; Biol Trace Elem Res 199 (2021) 2552–2564; Indian Journal of Microbiology 59 (2019), 379-382.
  2. Overall, the discussion section is weak and substantial improvement in the context of the quantitative data-based discussion to justify the significance of present finds with the citations.
  3. Please add one illustration to the summary of the present study along with the mechanism of antimicrobial action of biosynthesis NPs hydrogel.
  4. Few Figure’s quality should be improved.

About the comment:

“1.     Introduction, please add one paragraph on the advantages of nanoparticles (NPs) as antimicrobial agents, the suitability of hydrogel-based NPs synthesis and application or green synthesis approaches, how silver is suitable over other NPs such as gold, and the mechanism of NP’s action against pathogens, etc. RSC advances 6 (2016), 86808-86816; Biol Trace Elem Res 199 (2021) 2552–2564; Indian Journal of Microbiology 59 (2019), 379-382.”

Thank you for your suggestion. To highlight the advantages of bioAgNP incorporated into the hydrogel we developed, we added the following paragraphs to the introduction section:

“Among the nanometals, silver nanoparticles stand out, since they show action against several bacteria, in addition to being tested in combination with other antimicrobials in order to reduce toxicity and combat the emergence of bacterial resistance [23]. Firstly, because they have a broad spectrum of antimicrobial action, and can be incorporated into various products in our daily lives [24,25]. Among metals, silver has a great tendency to reduction (reduction potential equal to 0.80 V) compared to metals such as Cu (0.34 V), Fe (-0.44 V) and Zn (-0.76 V ) [26]; the reduction potential of these metals also con-tributes to their biological activity, since they bind to electron-donating microbial molecules and consequently interfere with the cellular function of the microorganism [27], but it is important to highlight that size, morphology, zeta potential, and composition are factors that also strongly influence the antimicrobial potential of nanometals [24].”

“There are commercialized products, including dressings and ointments for topical treatment, which contain chemically synthesized AgNPs in their composition [25], whose synthetic route uses toxic reducing agents and stabilizers [24]. On the other hand, bioAgNP are sustainable, since they are produced by “green nanotechnology” that does not require chemical reagents because the reducing and stabilizing properties of biological entities are used [24]. In addition, biomolecules confer high stability to silver nanoparticles by preventing their aggregation [28]. Another advantage is that the biological synthesis is more studied for AgNP than other metals, so that the bioAgNP production process has already been validated [24], as the F. oxysporum-bioAgNP used in the present study which have their mechanism of synthesis [29] and antimicrobial action investigated [20,22], which are important elements to justify the incorporation of these nanoparticles in hydrogel formulation presented in this study.”

About the comment:

“2.     Overall, the discussion section is weak and substantial improvement in the context of the quantitative data-based discussion to justify the significance of present finds with the citations.”

Thank you for pointing this out. We restructured the discussion section to be more in depth by exploring the results found in our study and relating them to the literature.  We improved the presentation of the nanoparticles that we incorporated into the hydrogel, since it is a well-studied and validated active by our research group. We believe that the discussion is better now, as it shows the importance of using this specific nanoparticle in the development of the present hydrogel that proved to be effective in preventing infection in open wounds, without harming healing and without causing bioaccumulation.

About the comment:

“3.     Please add one illustration to the summary of the present study along with the mechanism of antimicrobial action of biosynthesis NPs hydrogel.”

The development of a medicine involves steps and it can take years. It starts with the tests of effectiveness and mechanism of action of its pure active, as we have already done. The antimicrobial mechanism of these nanoparticles was previously investigated by our research group (*Scandorieiro et al., 2022; **Scandorieiro et al., 2023), and we presented such results in the discussion in the part we highlighted below. After that we incorporated this active into a pharmaceutical form and tested its effectiveness, this is the step presented in this article. The aim of the study we presented in the current article was to incorporate bioAgNP in a topical formulation and to investigate its in vivo effect in the treatment of open wounds, in terms of prevention of microbial infection, effect on the healing process and silver accumulation in the organs. According to the objective of the article, the main results were presented in the graphic abstract. We have added the paragraph below to clarify about the mechanism of antimicrobial action of F. oxysporum-bio-AgNP.

“Since F. oxysporum-bioAgNP are well-characterized, their antimicrobial activity has been studied by our research group; they showed activity against bacterial pathogens isolated from infected wounds, such as Staphylococcus aureus, Escherichia coli and Pseudomonas aeruginosa [20,21,44]. The mechanism by which silver nanoparticles exert their antibacterial activity is not yet fully understood; however, Scandorieiro et al. [20] showed that F. oxysporum-bioAgNP involves damage to the cell wall and cytoplasmic membrane, since disruption of these structures and cytoplasmic molecule leakage were identified by electron microscopy and spectrometry respectively. In addition, the antibacterial effect of these nanoparticles is due to oxidative stress confirmed by increased reactive oxygen species and lipid peroxidation. Other studies indicate that silver nanoparticles increase cell membrane permeability, inactivate enzymes, interfere with intracellular ATP levels, cause DNA damage and induce the formation of reactive oxygen species [45–49].”

* Scandorieiro, S.; Rodrigues, B.C.D.; Nishio, E.K.; Panagio, L.A.; de Oliveira, A.G.; Durán, N.; Nakazato, G.; Kobayashi, R.K.T. Biogenic Silver Nanoparticles Strategically Combined With Origanum Vulgare Derivatives: Anti-bacterial Mechanism of Action and Effect on Multidrug-Resistant Strains. Front Microbiol 2022, 13, doi:10.3389/fmicb.2022.842600

** Scandorieiro, S.; Teixeira, F.M.M.B.; Nogueira, M.C.L.; Panagio, L.A.; de Oliveira, A.G.; Durán, N.; Nakazato, G.; Kobayashi, R.K.T. Antibiofilm Effect of Biogenic Silver Nanoparticles Combined with Oregano Derivatives against Carbapenem-Resistant Klebsiella Pneumoniae. Antibiotics 2023, 12, 756, doi:10.3390/antibiotics12040756.

About the comment:

“4.     Few Figure’s quality should be improved.”

Thanks for suggestion. We improved quality of some figures, including figures 1B, 1C, 2A, 2B, and 2C. We believe they are better now.

Round 2

Reviewer 2 Report

  • All my questions were well answered and I agree to accept in present form.

  •  
  •  

Reviewer 3 Report

Accept